# The Interplay of Hippocampus and Ventromedial Prefrontal Cortex in Memory-Based Decision Making

**DOI:** 10.3390/brainsci7010004

**Published:** 2016-12-29

**Authors:** Regina A. Weilbächer, Sebastian Gluth

**Affiliations:** Department of Psychology, University of Basel, Basel 4055, Switzerland; sebastian.gluth@unibas.ch

**Keywords:** hippocampus, prefrontal cortex, episodic memory, value-based decision making

## Abstract

Episodic memory and value-based decision making are two central and intensively studied research domains in cognitive neuroscience, but we are just beginning to understand how they interact to enable memory-based decisions. The two brain regions that have been associated with episodic memory and value-based decision making are the hippocampus and the ventromedial prefrontal cortex, respectively. In this review article, we first give an overview of these brain–behavior associations and then focus on the mechanisms of potential interactions between the hippocampus and ventromedial prefrontal cortex that have been proposed and tested in recent neuroimaging studies. Based on those possible interactions, we discuss several directions for future research on the neural and cognitive foundations of memory-based decision making.

## 1. Introduction

Without a doubt, episodic memory and value-based decision making are amongst the most widely studied psychological constructs. Thus, when entering either of them as search terms in the research database PubMed [1], one obtains over 8000 results each. On the contrary, the combined term “memory-based decision making” produces only 77 results, with the great majority of them (i.e., 90%) dating not further back than the year 2000. This just reflects the novelty and relative under-appreciation of this specific research topic. In fact, the two research fields of episodic memory and value-based decision making have often adopted different research methods and developed separate conceptual definitions.

In our every-day life, memory- and choice-processes are often intertwined such as when choosing between chocolate bars in the supermarket and recalling past experiences [2]. Accordingly, in this review, we argue that even though memory and decision making can be understood as distinct processes, studying their interactions is a promising agenda for current and future research. We highlight recent advances in this regard and discuss how the existing knowledge about the neural mechanisms of memory, decision making, and their combination lead to novel predictions and research hypotheses. In particular, we will focus on two brain regions, the ventromedial prefrontal cortex (vmPFC) and the hippocampus (HPC), that appear to play crucial roles in memory encoding, consolidation and retrieval processes as well as in value-based decision making.

In order to have a common definitional ground and to avoid misunderstandings, let us define the central terms used in this article: episodic memory, value-based decision making, and memory-based decision making. *Episodic memory* is part of the declarative memory system (i.e., memory that can be stated) together with semantic memory (or memory for facts). Episodic memory is the memory for episodes, which means that when we recall a specific episode from the past, e.g., our first lecture at university, we recall not only what happened but also the specific context (in time and space) in which the event took place. *Value-based decisions* are decisions about what we like, want, or prefer, that is, choices based on the subjective value that the decision maker attributes to the available choice options. Value-based decisions are often distinguished from perceptual decisions, as the latter but not the former can be identified as being objectively correct or incorrect. Finally, *memory-based decision making* refers to the process of making decisions that require memory retrieval. For instance, when your friend is ordering Italian food and asks you to make your choice immediately (without presenting you the menu), you have to retrieve potential options from your memory and choose among those memory items. Importantly, not every value-based decision is a memory-based decision (at least according to our definition): when you have the menu at hand, there is no need to retrieve potential options from memory (although identifying your favorite Italian dish might still require episodic memory; cf. [2]).

This review article is structured as follows: first, we will summarize the literature of research on the role of HPC in episodic memory; second, we will give a similar overview with respect to vmPFC and value-based decision making; third, we will present recent work on the interplay of these two brain regions in both memory and decision making; fourth and last, we will propose research questions and hypotheses for future research in the field(s) that derive from our current knowledge.

## 2. The Central Role of the Hippocampus in Episodic Memory

The HPC comprises distinct sub-regions (i.e., CA1 to CA3, dentate gyrus and subiculum) and is part of the hippocampal formation, which also includes the parahippocampal, perirhinal and entorhinal cortices [3]. Figure 1 gives an overview of the input and output pathways of the hippocampal formation as described in [4]. The role of HPC in encoding, consolidation and retrieval of declarative long-term memory, especially episodic memory, is well established. Important contributions to our understanding of the HPC have been made by case studies such as patient H.M., who suffered from severe anterograde amnesia for facts and events (declarative memory) after bilateral removal of large parts of the medial temporal lobe, including HPC [5] (implicit long-term memory was not affected, as H.M. was able to learn procedural skills). Another relevant case study is patient R.B. who suffered from anterograde amnesia after a lesion of hippocampal CA1 region [6]. Additional evidence of the hippocampal role in episodic memory provide case studies of three children who suffered brain injuries at several ages (birth, four years old, and nine years old) in which HPC was affected [7]. These case studies showed that hippocampal lesion affected episodic but not semantic memory significantly.

More recent investigations suggest distinct roles of the different parts of the hippocampal formation for recognition memory [8,9]. Recognition memory is a form of episodic memory and is typically assessed by a task in which participants see different options and have to indicate which of them they encountered previously (old–new distinction). According to the role distinction view of different parts of the hippocampal formation, HPC appears to be mainly involved in creating associations between objects and places (source-based associative memory, a process termed recollection), the Parahippocampal cortex in place memory and the Perirhinal cortex in object familiarity (item-based non-associative memory) [8]. However, other results show a less clear localization of the two processes of familiarity and recollection, and a simple dichotomy of them may be too simplistic [10]. In fact, some researchers argue that the methods used to distinguish between recollection and familiarity instead separate between strong and weak memories, and higher HPC activity is associated with strong memories. This account is supported by the nonlinear relationship between hippocampal functional magnetic resonance imaging (fMRI) activation and memory strength [11]. Memory strength is defined as the rate of responses where participants clearly state that they remembered an item or indicate the source of the information. Therefore, a high response rate is associated with high memory strength. The nonlinear relationship mentioned above has been reported to be typical for the fMRI activation pattern (see Figure 2) and can be described as follows: during encoding as well as retrieval of weak memories, fMRI activation is constantly low, while during the linear increase of memory strength (leading to strong memories), the fMRI activation increases steeply [11].

During encoding [12] and retrieval [13] of long-term memories, we rely on working memory. Therefore, consolidation is the only “purely” long-term memory process. Concerning declarative memory consolidation and the role of HPC in it, the Multiple Trace Theory (MTT) [14,15] has been put forward and subsequently updated as transformation hypothesis [16]. MTT states that HPC is always required for encoding as well as recall of episodic memories. During retrieval, a new trace element is added and the memory is strengthened. According to MTT, contextual information will always stay in the HPC. Over time, however, this contextual information can be lost and the memory may be transformed into more generic (semantic) memory stored in the neocortex so that it is not affected by HPC lesions anymore [15,16]. MTT seems to account better for the process of declarative memory consolidation than previous theories such as cognitive map (CM) or the system consolidation model (SC) [15]: the CM theory [17] sees the HPC involved in the creation of an allocentric spatial representation of the environment that is independent from the position of the observer. Contextual information is based upon this representation and as the context is a constituent part of episodes, HPC is important for the encoding, consolidation and retrieval of episodic memory. The central difference to MTT is that the CM theory does not differentiate between remote and recent memories. The SC model [18] assumes a process of prolonged consolidation, which is only temporally HPC dependent. Over time, however, neocortical regions store long-term memories and also mediate their retrieval. Contrary to MTT, also contextual information, like any other declarative long-term memory, becomes HPC independent over time. Additionally, SC does not differentiate between episodic and semantic memory.

The work we have outlined above focused on human studies. However, episodic-like memory in animals has also been investigated in recent years. Compared to human studies, animal models allow a first and simplified understanding of functionally complex cognitive processes as well as neuropsychiatric and neurodegenerative diseases in simpler models. Animals cannot tell us what events they remember, but we can derive their knowledge of past events from their behavior [19]. An example of an episodic-like memory task is the WWWhen [20,21] (what, where and when; for an overview see [19]). In the WWWhen task, animals perform a three-trial object exploration task. During this task, long-term memory for different objects (what), their spatial location (where) and their order of presentation (when) is assessed [20]. In other animal studies, hippocampal place cells, in most cases pyramidal CA1 cells, have been suggested to underlie spatial navigation in rats [22,23,24]. Spatial knowledge is consolidated via spontaneous recurrence of HPC place cell activity during slow-wave sleep (SWS) [25] in animals [26] and humans [27]. Such neural replay has also been reported in awake states, for instance, when rodents pause in exploring a maze [28,29]. Neural replay can be either forward or reverse [22]. Reverse replay has been suggested to be a learning mechanism while forward replay could be crucial for memory retrieval and future paths planning [22,30].

In summary, the HPC is central to guiding self-referenced navigation as well as to supporting the encoding, consolidation and retrieval of declarative episodic memory [31].

## 3. The Ventromedial Prefrontal Cortex Is Essential for Value-Based Decisions

Before we review the extent of the literature on the critical role of the vmPFC for decision making, we provide a brief overview of the anatomy and connectivity of this cortical region. The vmPFC is not an anatomically distinct area, as it is not restricted to specific Brodmann areas (BA) or standard MRI coordinates. The vmPFC can be subdivided into caudal (posterior) and rostral (anterior) parts. The caudal vmPFC includes BA 25 and 32 (primarily subgenual cingulate bundle) and caudal BA 14 [32]. The rostral vmPFC, although not explicitly described according to BA in [32], could include BA 10, 11 and 32 (in humans), as described in [33]. However, researchers have also delineated this area on the basis of anatomical and functional connectivity analyses performed with (f)MRI data. Thus, vmPFC could also be seen as a cluster resulting from parcellation, where the fMRI signal changes for a specific construct, such as rewards [34]. Figure 3 gives an overview of the principal connections from and to the vmPFC (as suggested by [32]). The vmPFC has connections to and receives input from the dorsal, anterior and ventrolateral prefrontal cortex, the dorsal anterior cingulate cortex, as well as the insula. Unidirectional connections to the vmPFC come from the hypothalamus and HPC, while output regions are mainly the shell of the nucleus accumbens and the amygdala. Additionally, vmPFC is bidirectionally connected to the entorhinal and perirhinal cortices [35,36,37,38], which are part of the hippocampal formation and are, in turn, connected to the HPC (cf. Figure 1).

A wealth of research in cognitive neuroscience over the last three decades has established the vmPFC as a core brain structure for value-based decision making [39,40,41]. First of all, lesions to the vmPFC lead to various decrements in decision making abilities, including learning from reward and punishment [42], making transitive choices [43], making future-oriented decisions [44], or directing attention towards reward-predicting information [45]. Neuroimaging studies have established a remarkably strong coupling between activity in vmPFC (and ventral striatum) and the subjective value of available choice options or any presented stimuli in general [46,47,48,49] (for reviews/meta-analyses, see [41,50]). This has led to the neuroeconomic proposal of a common currency for the vmPFC: in order to allow value-based comparisons between any form of putatively incommensurable choice options (e.g., a choice between buying a car or having a trip around the world), the subjective value of each option is translated into the common currency of vmPFC activation, so that the option with the highest subjective value can be identified by means of the neural vmPFC signal [51]. Interestingly, a recent review article suggests that the vmPFC may receive memory-related information from the HPC in order to estimate the subjective value of options that have been experienced in the past [2]. Importantly, the authors of this review article argue that this HPC-vmPFC communication should underlie value-based decision making, even when choice options are directly visible and do not have to be (explicitly) retrieved from memory.

In contrast to the widely accepted view of the vmPFC as representing subjective value, it is still a matter of debate whether the vmPFC also processes the decision itself. In other words, does the vmPFC represent the value of options and transfers this information to downstream areas that select the best option via a dynamic comparison process, or does this dynamic comparison process take place in the vmPFC itself? Some studies, including single-unit recording studies in non-human primates, have identified activation patterns in vmPFC that are best accounted for by assuming comparison-like mechanisms such as evidence accumulation or mutual inhibition [52,53,54]. Other work points to the dorsomedial PFC, including the anterior cingulate cortex and the pre-supplementary motor area (pre-SMA), as a potential downstream area that receives information from the vmPFC in order to select among the available choice options [55,56,57]. The diversity of choice-related impairments that are caused by lesions to the vmPFC (see above) seems to favor the former account. On the other hand, it is well established that decision making is not processed in a strictly serial manner by the brain (i.e., identification of choice options → representation of values → comparison of values → execution of an action), and that even purportedly motor-related areas such as pre-SMA or the primary motor cortex play an active role during the choice process [56,58,59,60].

On a related note, a large number of behavioral studies [61,62,63] and a few neuroimaging experiments [64,65,66,67] have shown converging evidence that valuation and choice processes cannot be separated from each other. This is because the relative choice probabilities of two options depend on the presence of other options in the choice set, a phenomenon denoted as context effect. Such effects imply that the brain does not assess option values independently from each other before starting the choice process, but that valuation and choice mechanisms must be intertwined. With respect to the vmPFC’s role in value-based decision making, it remains open whether context effects lend further evidence for a common valuation and choice mechanism within vmPFC, or whether bi-directional interactions between vmPFC and downstream areas could also account for them. Figure 4 provides a schematic overview of the core features of value-based decision making and the associated brain regions.

## 4. Hippocampal-Prefrontal Interactions in Episodic Memory and Decision Making

In recent years, researchers have started to investigate the interplay of HPC and vmPFC in several related contexts, such as retrieval-mediated learning [68], learning and choosing based on conceptual (episodic) knowledge [69], deliberative decision-making [70] or memory-based preferential choices [71]. Although more and more research focuses on the connections between these two regions, the neural mechanism underlying the coordinated action of HPC and PFC are still unclear [72]. In this chapter, we present two methods used to measure the neural mechanisms underlying the coupling between HPC and PFC. The first method employs coherence of theta- and gamma band oscillations, and the second method performs dynamic causal modeling of fMRI data. We then present theories and frameworks that were inspired by the results obtained from these connectivity studies. They include the differentiation between HPC and PFC concerning their roles as well as memory as a process of schema creation vs. integration.

The first method used to investigate the coupling between HPC and PFC is the phase coherence of oscillations across specific frequency ranges. Those oscillations are usually measured with magneto encephalography (MEG) and local field potentials (LFP) in humans and non-human animals, respectively. Hippocampal theta-band (5–10 Hz in [73] or 4–8 Hz in [74]) oscillations [75] have been reported to be associated with (spatial) learning and memory consolidation in rats [73,76]. Theta-band synchrony with prefrontal regions has been investigated in several rodent studies concerning memory (for a review, see [77]) as well as in human decision making [74]. Hippocampal–prefrontal gamma-band (low: 35–55 Hz, high: 65–90 Hz) oscillations synchrony has been also linked to spatial learning and memory [78]. Recent research indicates that gamma and theta oscillations might not be independent from each other. In fact, gamma oscillations recorded in neocortical areas appear to be biased by hippocampal theta oscillations via cross-frequency coupling [79]. The authors explain this biasing effect by the “reciprocal information transfer” [79]: The recipient brain structure temporally biases activity in the source structure, and, as a consequence, the recipient structure can receive information more effectively. However, a demonstration of causality between oscillations coherence and cognitive processes (e.g., memory or decision making) is still missing [72]. Consequently, the measurement of oscillation coherence is a promising method but needs further specifications in order to provide evidence for a causal role of those oscillations in mediating cognitive processes.

The second method investigating HPF-PFC coupling has been used in a recent fMRI study [71], where Gluth and colleagues studied how people make decisions about options that are not directly visible but have to be retrieved from memory. Participants learned associations between choice options (food snacks) and locations on the screen and then decided between two options based on their locations. The authors found that memory exerts a bias on value-based decisions, that is, options that were better remembered were preferred even if their subjective value was comparatively low. Dynamic Causal Modeling (DCM) [80] of the fMRI data, a technique that allows measuring the extent and direction of effective connectivity between distant brain regions, revealed that the coupling from HPC to vmPFC was not only important for processing memory-based decisions in general but also for mediating the memory bias. Figure 5 shows the most likely DCM network of effective connectivity between HPC and vmPFC. The strength of HPC-vmPFC coupling was dependent on whether participants chose the better remembered snack or not.

Based on the findings from these studies on the functional neural coupling between HPC and PFC, different theories and frameworks have been put forward. First of all, Shin and Jadhav [72] conclude that the bidirectional flow of memory-related information between HPC and PFC supports memory formation (consolidation), contextual memory retrieval and memory-guided decisions. Specifically, the insights from neural connectivity analyses seem to strengthen the hypothesis of a role differentiation between PFC and HPC, where HPC is mainly involved in memory encoding while the PFC is more engaged during memory retrieval [81,82]. Accordingly, during the consolidation process, the vmPFC is supposed to take over the role of HPC making the stored events more accessible to the PFC [83,84]. This view clearly opposes the previously presented MTT, stating that contextual information will always be kept in the HPC [16] but supports more the account of SC [15]. Moreover, a recent review [85] suggests that vmPFC is similar to a control instance, as it selects the appropriate memory for a specific context, and then controls the retrieval of the detailed memories in the HPC (context-guided retrieval). This explanation is reminiscent of the classical view of PFC as “central executive”. In addition, vmPFC may not be crucial only at the end of consolidation but already during earlier stages of learning, e.g., in the development of schemata. According to Piaget [86], a schema is a structured mental representation of related associations. Preston and Eichenbaum [85] reviewed the interplay of HPC and PFC in memory concluding that vmPFC and HPC interact during schema formation (encoding), consolidation and expression (retrieval). Furthermore, it has been suggested that only new events overlapping with previous experiences need the interplay of HPC and vmPFC. This is the case for items learned according to the association inference paradigm: in this paradigm, a first association between two items is learned (A-B, e.g., the pair ball-hat), and then a second association is learned that includes a known and a new item (B-C, e.g., the pair hat-fork). The schema A-B-C refers to the fact that the three items ball-hat-fork belong to the same group. During the stage of schema formation, the strategic roles of vmPFC and of the vmPFC-PC coupling come into play. After having learned the schema, subjects can infer that A-C (i.e., ball-fork) also belongs to the same group although they were never shown together. In contrast, if events do not overlap, HPC may integrate the memories predominantly alone (resulting in a lower vmPFC-HPC coupling). This would be the case when, for example, we learn two completely new and independent associations like A-B and C-D, for which we have no previous knowledge. The previously described processes can be linked to the classical concepts of assimilation and accommodation [85] introduced by Piaget, who also shaped the definition of a schema [86].

Finally, also derived from findings showing evidence for a neural mechanism underlying the connectivity between HPC and PFC, Wang and colleagues proposed a framework for understanding cognitive and/or behavioral choices [87]. The framework is called Covert Rapid Action-Memory Simulation (CRAMS) [87] and states that covert memory processing of HPC interacts with action-generation processing of PFC in order to arrive at memory-guided choices with little effort. The term “covert” indicates that this process can take place without conscious awareness (implicitly and automatically) and this is consequently the reason why the process is also “rapid”. This framework aims to explain the mechanism underlying difficult decisions, that is, decisions where the options appear to have the same outcome value. In those situations where a response conflict arises, PFC provides the HPC with possible plans for action via CRAMS. In other words, first the possible actions are simulated (lateral PFC), and then covert memory is retrieved (HPC) and finally evaluated (medial PFC). This process is repeated until a goal threshold is reached. Afterwards, the action can be performed, and a choice is made (motor systems). Interestingly, the CRAMS model is in accordance with the above mentioned notion that HPC-dependent memory retrieval should be relevant even in value-based decisions that do not require an explicit recall of information from the past [2]. However, it is widely acknowledged that not only PFC but also HPC contributes to imaging future events [88,89,90,91]. As far as we know, the CRAMS framework does not account for those findings. Indeed, the fact that HPC may simulate the past and/or the future could represent an important extension to this framework. Additionally, the CRAMS model is currently only a theoretical framework and needs specific empirical testing in order to be able to relate a specific region to a distinct function.

## 5. Open Questions and Tentative Predictions for Future Studies on the Interplay of Value-Based Decisions Making and Episodic Memory

Even though the findings of HPC-vmPFC interactions outlined in the previous chapter are remarkable and promising, we are still far away from having a clear picture of how exactly they work together in enabling memory-based decision making. So far, the studies differ greatly with respect to their research questions, methodologies, and interpretations of the data. In fact, researchers even refer to different things when they speak of “memory-based decisions”. In contrast to our definition (see Section 1), this term is sometimes used to refer to decisions about memory, that is, whether a currently presented stimulus has been encountered before or not (old vs. new) [92]. In the following, however, we will focus on value-based decisions that require the retrieval of information from memory [71], that is, decisions about (memory-based) preferences and not about memory itself (for a definition of the different concepts see the Section 1). The goal of the present chapter is to identify relevant open questions and to make some tentative predictions of how HPC and vmPFC interact with respect to memory-based decision making by a critical appraisal of what we already know and what we do not know in this regard.

### 5.1. Question 1: What Is the Neural Code That Underlies Memory-Based Decisions?

As outlined in the previous chapter, there is ample evidence that HPC-vmPFC interactions rely on the coherence of low-frequency (i.e., theta-band) long-range neural oscillations. The study by Guitart-Masip and colleagues [74] indicates that such a synchronization of neural signals in the theta-band may also underlie memory-based decisions. However, the task paradigm employed in this study is not a pure episodic memory task but has typical reinforcement learning features. Hence, it remains open whether theta-band oscillations between HPC and vmPFC are critical in decisions that are based on retrieving episodic memory content, and also whether these oscillations are generally beneficial for making accurate decisions [69,74] or whether they might bias decisions toward better memorized choice options [71]. The use of the MEG technique together with source reconstruction methods will be crucial for answering these questions.

Additionally, Guitart-Masip and colleagues [74] reported no significant effects of reward, punishment or their interaction on the low-frequency coupling between HPC and vmPFC. This result is not in line with the DCM results of Gluth et al. [71] and is also difficult to reconcile with accumulating evidence for value-based signals in HPC (see Question 4 below). Again, such discrepancies might be due to the fact that different tasks do or do not contain reinforcement learning elements. Hence, it will be important to employ similar experimental paradigms with different neuroimaging techniques such as fMRI and MEG in the future.

### 5.2. Question 2: How Can We Bring Schema-Related and Choice-Related HPC–vmPFC Interactions Together?

Hippocampal-vmPFC interactions seem to lie at the heart of both enabling complex inferential memory associations [68] as well as mediating value-based decisions from memory [71], but how can we bring these two findings together? A possible answer could be that an increased HPC–vmPFC coupling enables a richer representation of (choice-relevant) past events so that decision making is altered on a qualitative level. For example, if a currently visible cue A (e.g., a deck of cards) triggers only a direct association with another event B (e.g., gambling in a casino) that has been linked to A in the past, the decision to engage in an action might differ as compared to when an (HPC–vmPFC coupling dependent) indirect association from A via B to C (e.g., the loss of money) is also triggered.

Importantly, the associations created in the example above are of personal value to the participants. In the study of Zeithamova and colleagues [68], however, the association task included different images with assumed neutral valence, such as common objects and outdoor scenes. An additional valence-rating task could provide information about whether the images are really perceived as neutral. Alternatively, pictures from the International Affective Picture System (IAPS) [93], for which arousal and valence information is available, could be used in such memory- and schema-based task. The general point that we want to make here is that “memory researchers” should be aware of the fact that the vmPFC, one of the brain regions they are (currently) interested in, is very sensitive to value-related information (which might also be true for the HPC; see Question 4 below). This has to be taken into account to avoid confounding effects and misinterpretations. On the other hand, “decision-making researchers” should be aware of the fact that different choice options might easier or harder to store into and retrieve from memory (e.g., [71]).

### 5.3. Question 3: What Is the Direction of HPC–vmPFC Connectivity?

For the connectivity analysis in [71], DCM was applied which (in contrast to methods such as psycho-physiological interactions) allows identifying the direction of the information flow from one brain region to another. The authors tested different potential circuits with connections from HPC to vmPFC, from vmPFC to HPC, and bidirectional connections, and found the first circuit (i.e., from HPC to vmPFC) to provide the best account of the data. Interestingly, context-guided retrieval of memory as in the processing of mental schemata has been hypothesized to instead rely on the vmPFC impacting on activation in the HPC [85]. Therefore, it is tempting to speculate that the direction of HPC-vmPFC coupling can dissociate between the processes of retrieving information from memory to guide decision making and the process of constructing complex memory representations. Accordingly, it will be important for future studies to apply methods such as DCM (which is not only applicable to fMRI but also to MEG) that allow identifying in which way HPC and vmPFC communicate with each other.

### 5.4. Question 4: Does the HPC Represent Subjective Value?

The common view is that the vmPFC (perhaps together with the ventral striatum) represents subjective value, while HPC is crucial for episodic memory. Some studies, however, found an activation in HPC during the encoding of subjective value [47,71,94,95,96]. A question that arises here is how the HPC can access value information. Two possibilities have been proposed [94]: on the one hand, the HPC may receive these information via its connections with vmPFC/ventral striatum and is thus only indirectly linked to value computation; alternatively, the HPC might play a direct role in value processing. Importantly, most studies that reported strong correlates of subjective value in HPC employed an experimental paradigm that requires people or animals to rely on spatial memory for making good decisions [71,95,97,98]. Hence, we speculate that a putative direct value-coding function of HPC may be restricted to spatial memory demands. This hypothesis could be tested by modulating spatial vs. non-spatial episodic memory demands within the same task.

On a more general note, we want to repeat our appeal to researchers in the two fields of episodic memory and value-based decision making that they should be aware of the fact that they are studying interrelated constructs that rely on interrelated brain regions. We hope that reviews such as the present article will help to sensitize researchers of both fields that their findings may sometimes be easier and better explained by referring to these relationships of memory and decision making rather than by strictly staying within the theories of a single domain of research.

## 6. Conclusions

In the present review article, we have reviewed the classical views concerning the roles of HPC and vmPFC in episodic memory and value-based decision making, respectively, and we highlighted various methods and theories concerning the interplay of HPC and vmPFC. Based on our emerging knowledge about the cross-talk of these brain areas, we proposed several directions for future research. In our opinion, designing studies that address those questions will lead to a clearer picture of the reasons why and the circumstances in which HPC and vmPFC interact in processing memory and decision making. Methods that allow joint measurements of HPC and vmPFC activity as well as their connectivity profile, such as MEG and fMRI, will be of extraordinary importance for this endeavor.

## Figures and Tables

**Figure 1 brainsci-07-00004-f001:**
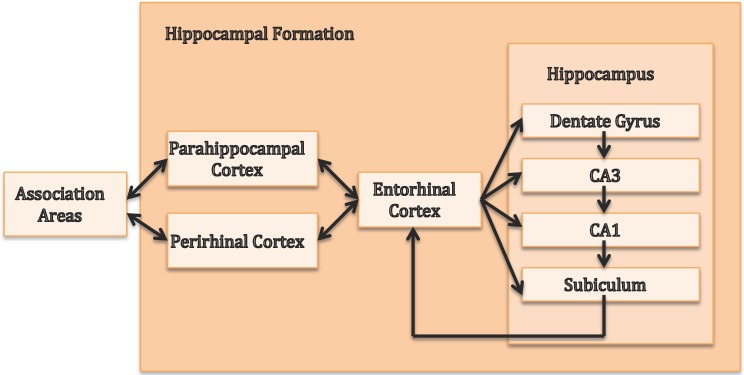
Input and output pathways of the hippocampal formation (adapted from [4]).

**Figure 2 brainsci-07-00004-f002:**
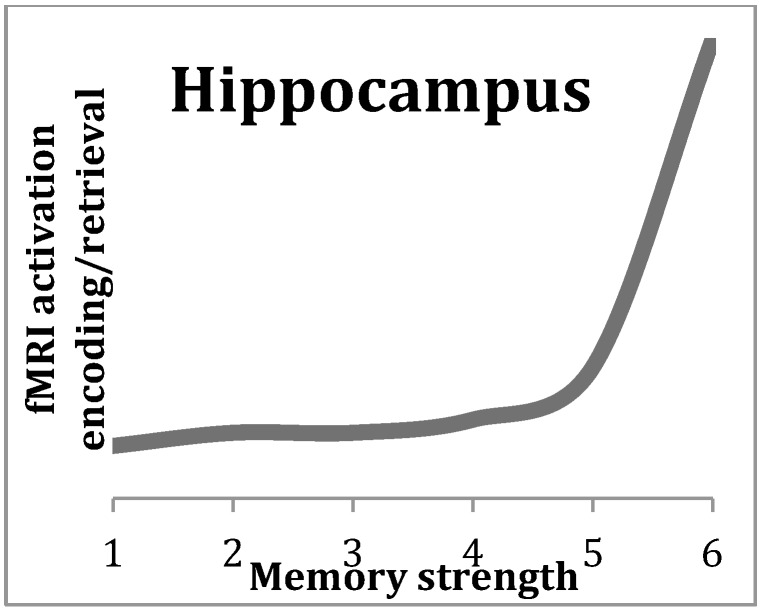
Nonlinear relationship between functional magnetic resonance imaging (fMRI) activation (ordinate) and memory strength (abscissa) in the hippocampus (HPC). The pattern is equal during encoding as well as during retrieval. The memory strength in this example ranges from 1 (weak) to 6 (strong). The figure shows that weak to middle strong (1–5) memories show a constant fMRI activation, where very strong (6) memories are associated with an extremely high fMRI activation in the HPC (according to [11]).

**Figure 3 brainsci-07-00004-f003:**
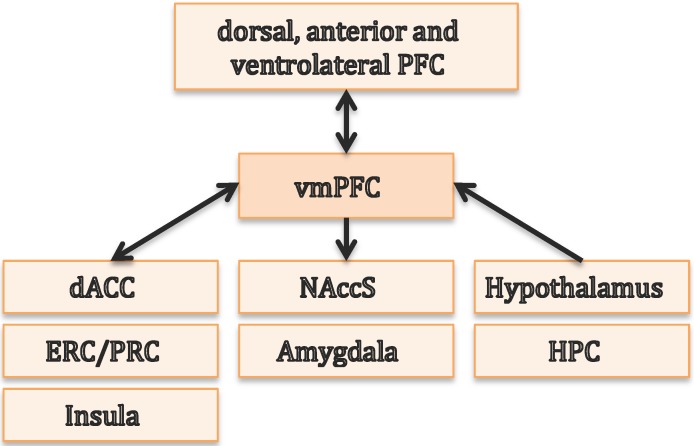
Principal connections from and to the ventromedial prefrontal cortex (vmPFC) adapted from [32]. dACC = dorsal anterior cingulate cortex; ERC = entorhinal cortex; PRC = perirhinal cortex; NAccS = nucleus accumbens shell; HPC = hippocampus.

**Figure 4 brainsci-07-00004-f004:**
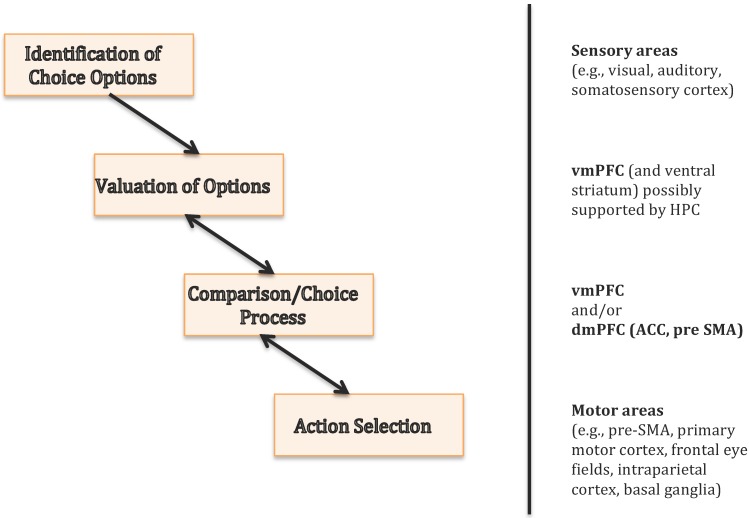
Value-based decision making in the human brain. The central features of a value-based decision are outlined on the **left**; associated brain regions are listed on the **right**. The bidirectional arrows between valuation and choice as well as choice and action selection indicate that these mechanisms are not processed in a strictly serial and independent manner and may even be computed within a single brain region. vmPFC = ventromedial prefrontal cortex, dmPFC = dorsomedial prefrontal cortex, ACC = anterior cingulate cortex, HPC = hippocampus, pre-SMA = pre-supplementary motor area.

**Figure 5 brainsci-07-00004-f005:**
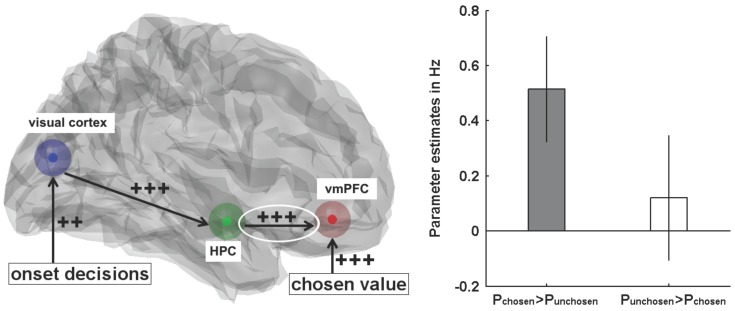
Effective connectivity in preferential choices from memory. **Left**: the most likely Dynamic Causal Model with connection strengths (‘‘++’’ = positive at *p* < 0.01; ‘‘+++’’ = positive at *p* < 0.001); **right**: the connection from hippocampus to ventromedial prefrontal cortex (circled) was only significantly positive when participants chose the better remembered snack. Error bars represent 95% confidence intervals (adapted with permission from [71]).

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
