# Peer review of "The Interplay of Hippocampus and Ventromedial Prefrontal Cortex in Memory-Based Decision Making"

_brainsci, 2016, doi:10.3390/brainsci7010004_

Round 1

Reviewer 1 Report

This review presents evidence to support the view that episodic memory and decision making share neural substrates, i.e the hippocampus and ventromedial prefrontal cortex (vmPFC), and thus memory based decision making arises from the interplay between these two neural regions.

The authors begin by reviewing the literature concerning the separate roles of the hippocampus in episodic memory, and the vmPFC in decision making.  before discussing the importance of hippocampal-prefrontal interactions and finally present some questions concerning the nature of the interaction in effecting memory-based decisions.

The paper presents an interesting perspective, and synthesises the appropriate literature.  I have a few comments on the presentation of the inforamtion however which are detailed below

1. The authors refer to evidence on the role of the hippocampus is episodic memroy from patient HM, however it m,ight be worth also citing evidence from amnesitci patuents who sustained more selective hippocampal damage.  there are a number of such case studies in the literature.

2. In the second paragraph of page two,  the authors discuss the roel fo the hippocampus in recognition/recollection, however previously the authors sicussed episodic memory.  hence the realtionship between these different terms needs to be elaborated to avoid confusion.

3. on a related point a deficition of the term 'episodic memory' might be useful.

4. page 3.. the authors jumnp between episodic memroy in humansd and spatial navigation/place cells in rodents.  the relationship between these cogntiive processes between the two species would be helpful.  Also the authors may want to cite evidence of the role of the hippocampus in 'episodic-like' processes in animals.

5. A presentation fo the anatomical subdivisions of the prefrtonal cortex, and thier anatomical connections might be useful and would parallel the section on the hippocampus. 

minor points

line 56 the use of the terms hippocampus (HPC) and hippocampal formation (HF) might be confusiong for someone coming into the field without detailed background knowledge (eg. a student)

line 215 the use of the word 'he' to refer to the PFC is slightly odd, may be 'it' would be better

Author Response

Dear Reviewer

Please find attached the word .docx document containing our reply. 

Kind regards, 

Regina Weilbächer and Sebastian Gluth

Reviewer 2 Report

This is an interesting review of the current state of play of interactions between the hippocampus and ventromedial prefrontal cortex in modulating memory-based decision making. The manuscript is well-written and offers a timely perspective on an important, and poorly understood, branch of cognitive neuroscience. Overall, I enjoyed reading this review however, there are a number of issues I would like to see addressed.

1. The goal of this review is to provide an update on theories of HPC-vmPFC interactions in relation to memory-based decision making. Crucially, however, the authors do not provide a clear working definition of what they mean by “memory-based decision making”. We are given a definition late in the piece (page: 7) yet this construct seems thereafter to be equated with value-based decision making. How the authors operationalise this construct is critical for the overall message of the manuscript. As such, I would like to see a formal definition of memory-based decision making presented at the start of the review, along with how this differs from value-based decision making. It would further be helpful to comment on how the definition and therefore measurement of the construct potentially differs across studies, and what bearing this has on the results elicited.

2. Related to the above point, I would like to have seen more critical appraisal and synthesis of the disparate streams of research to highlight gaps/methodological failings, etc. and comment on where the field needs to go. The authors arguably do provide suggestions for future directions, yet the critical appraisal of the piece is somewhat lacking.

3. The authors provide a clear neuroanatomical delineation of the HPC on page: 2, yet the regions/Brodmann areas which make up the vmPFC are not described in this manner. It is important for the reader to appreciate what areas are subsumed under their conceptualisation of the vmPFC and potentially how this might diverge across studies.

4. Finally, on page: 7 the authors suggest that according to the CRAMS model, possible actions are simulated by the lateral PFC. I wonder how this model can account for the now well established hippocampal contribution to simulation and whether, again, such discrepancies across studies reflect divergent definitions or methods of measurement.

Author Response

(The authors gave the same response as above.)

Round 2

Reviewer 1 Report

The authors have satisfactorily addressed all the points I raised in my original review